# Ice-Templated Cellulose Nanofiber Filaments as a Reinforcement Material in Epoxy Composites

**DOI:** 10.3390/nano11020490

**Published:** 2021-02-15

**Authors:** Tuukka Nissilä, Jiayuan Wei, Shiyu Geng, Anita Teleman, Kristiina Oksman

**Affiliations:** 1Fibre and Particle Engineering Research Unit, Faculty of Technology, University of Oulu, FI-90014 Oulu, Finland; Tuukka.nissila@oulu.fi; 2Division of Materials Science, Department of Engineering Sciences and Mathematics, Luleå University of Technology, SE-97187 Luleå, Sweden; Jiayuan.wei@ltu.se (J.W.); shiyu.geng@ltu.se (S.G.); 3RISE Research Institutes of Sweden, SE-11428 Stockholm, Sweden; anita.teleman@rise.se; 4Mechanical & Industrial Engineering (MIE), University of Toronto, Toronto, ON M5S 3G8, Canada

**Keywords:** cellulose nanocomposite, ice-templating, interface, orientation, mechanical properties

## Abstract

Finding renewable alternatives to the commonly used reinforcement materials in composites is attracting a significant amount of research interest. Nanocellulose is a promising candidate owing to its wide availability and favorable properties such as high Young’s modulus. This study addressed the major problems inherent to cellulose nanocomposites, namely, controlling the fiber structure and obtaining a sufficient interfacial adhesion between nanocellulose and a non-hydrophilic matrix. Unidirectionally aligned cellulose nanofiber filament mats were obtained via ice-templating, and chemical vapor deposition was used to cover the filament surfaces with an aminosilane before impregnating the mats with a bio-epoxy resin. The process resulted in cellulose nanocomposites with an oriented structure and a strong fiber–matrix interface. Diffuse reflectance infrared Fourier transform and X-ray photoelectron spectroscopy studies revealed the presence of silane on the filaments. The improved interface, resulting from the surface treatment, was observable in electron microscopy images and was further confirmed by the significant increase in the tan delta peak temperature. The storage modulus of the matrix could be improved up to 2.5-fold with 18 wt% filament content and was significantly higher in the filament direction. Wide-angle X-ray scattering was used to study the orientation of cellulose nanofibers in the filament mats and the composites, and the corresponding orientation indices were 0.6 and 0.53, respectively, indicating a significant level of alignment.

## 1. Introduction

Nanocellulose is considered to be an ideal renewable alternative to the more commonly used reinforcement materials in composites [1]. It is widely available as it can be either produced from a variety of plant sources or synthesized by bacteria, and its mechanical properties are outstanding. The nanoscale fibers are more defect-free compared with natural fibers or microscopic native cellulose, and Young’s modulus has been estimated to be approximately 100 GPa [2], a value comparable to that of Kevlar [3]. The strength can be as high as several gigapascals [1]. The nanoscale size also contributes to the additional benefit of high specific surface area [4,5], which leads to a high amount of interfacial area between the fibers and the matrix material and thus to a potentially strong reinforcement effect.

One of the main challenges in cellulose nanocomposite processing is controlling the fiber arrangement inside the matrix material, especially when hydrophobic polymers are used. A widely utilized method is creating various continuous structures, such as filaments, nanopapers, and aerogels. The tendency of nanocellulose to form networks via strong fiber–fiber hydrogen bonds has been known since the earliest studies on cellulose nanocomposites [6,7]. It was shown that the mechanical properties of a polymer matrix could be significantly improved even with a low fraction of the nanoscale fibers. The remarkable effect was attributed to a percolation phenomenon, i.e., the formation of a continuous network structure that provides a stress transfer mechanism outperforming the theoretical predictions for the mechanical properties of short-fiber composites. Later, the network-forming tendency of nanocellulose has been used to prepare nanopapers that have then been impregnated with a polymeric resin to obtain composites [8,9,10]. The high specific surface area reported for nanopapers suggests that such fiber networks consist of well-individualized nanoscale fibers that provide effective reinforcement [4]. This approach has resulted in relatively high fiber contents and promising mechanical properties. However, the filling process is typically time-consuming, and often an organic solvent has to be used to facilitate the impregnation. Another means of controlling the organization of the nanocellulosic entities is the formation of single filaments similar to those traditionally used in the textile industry [11,12,13,14,15,16,17]. Dry spinning, wet spinning, and hydrodynamic alignment are some variations of the same approach and have all been successfully utilized in nanocellulose processing. The filaments obtained via the spinning process are commonly in the micrometer scale. Considering the application of such materials in composites, the reported average diameters ranging 6.8–250 µm [11,12,13,14,15,16,17] suggest that the full potential of having a nanoscale raw material has not yet been reached. A better reinforcement effect could be expected by finding ways to decrease the lateral dimension of the filaments.

Ice-templating is a method that can be used to turn a nanocellulose water suspension into a porous honeycomb-like structure [18,19,20]. The resulting aerogels are composed of aligned pores that run through the material in the direction of the ice crystal growth and can be used as preforms for composite materials by filling the structure with a polymer [21,22,23]. However, these kinds of honeycomb structures can only be impregnated in the pore direction, making the process time-consuming and impractical. In addition, the relatively low specific surface area of ice-templated aerogels suggests that the nanofibers have aggregated during the formation of the honeycomb, and again, the benefit of having a nanoreinforcement material is lost [18,22,24]. Threadlike nanocellulose filaments can be produced with the same approach by decreasing the fiber content of the suspension [25,26]. Instead of forming a self-standing monolithic honeycomb foam, the nanofibers assemble into thin filaments with a diameter as small as a few hundred nanometers. In turn, the filaments are arranged into a mat-like and partially interconnected material in which they are oriented along the freezing direction. This type of open structure can be impregnated with a liquid resin in the through-plane direction instead of filling the honeycomb pores gradually from one end to the other.

In addition to the difficulty of controlling the arrangement of the nanocellulose fibers, the interfacial adhesion between the hydrophilic reinforcement material and the most common thermoset resins tends to be poor [27]. Using an organosilane to treat the surface of the fibers improves the properties of cellulose and natural fiber/epoxy composites [28,29]. This is the case especially when an aminosilane, capable of forming covalent bonds with the resin, is used [29]. However, the commonly used solution-based silylation is not easily applicable to dried nanocellulose materials, because they tend to deform during the evaporation of the water-containing solvent. Chemical vapor deposition (CVD) is a more straightforward and efficient way to functionalize and hydrophobize cellulosic nanomaterials, such as cellulose nanofiber aerogels [23] and filaments [16]. In its simplest manifestation, the process consists of placing the cellulosic material and the liquid silane inside a closed container and vaporizing the chemical by heat and/or vacuum to cover the cellulosic surfaces. Any solvents or subsequent processing steps are not needed.

In order to control the assembly of the cellulose nanofibers into a macroscopic structure, we utilized ice-templating to prepare oriented filament networks (Figure 1). These nonwoven cellulose nanofiber (CNF) filaments were then used as a reinforcement material in composites by filling the preforms with a bio-epoxy resin via vacuum infusion. The filament surfaces were treated with an aminosilane using a simple lab-scale CVD process to improve the interfacial adhesion with the matrix material. This paper presents the method of preparing the nanocellulose filaments and the composites along with the results of microscopy, diffuse reflectance infrared Fourier transform (DRIFT) spectroscopy, X-ray photoelectron spectroscopy (XPS), wide-angle X-ray scattering (WAXS), and dynamic mechanical analysis (DMA) studies.

## 2. Materials and Methods

### 2.1. Materials

Bleached softwood sulfate pulp (Stora Enso, Oulu, Finland) was used as a CNF source. The chemical composition of the pulp was reported in our previous study: cellulose 96.3%, hemicellulose 2.4%, lignin 1.3%, and the crystallinity index 68% [30]. The pulp with a 2.0 wt% concentration was passed through an ultrafine grinder (Supermasscolloider MKCA 6-2J CE, Masuko Sangyo Co., Ltd., Kawaguchi, Japan) until a final gap of −90 μm relative to the initial contact point of the grinding stones was reached. Further fibrillation was achieved via microfluidization (Microfluidics M-110EH-30, Westwood, MA, USA). The material was passed twice through 200 and 200 µm chambers at 1000 bar, twice through 200 and 100 µm chambers at 1500 bar, and finally twice through 200 and 87 µm chambers at 1500 bar. Figure 2 shows field emission scanning electron microscope (FESEM) and transmission electron microscope (TEM) images of the resulting CNFs. The size distribution was calculated based on measuring the width of approximately 250 fibers from the TEM images. The width distribution was between 10 and 30 nm. Super Sap CLR (Entropy Resins, Hayward, CA, USA) bio-epoxy was used as the polymer matrix. The resin was mixed with Super Sap INH and CLX hardeners using a 100:19:19 (CLR:INH:CLX) mixing ratio based on weight. 3-Aminopropyltriethoxysilane (APTES, 98%, Sigma-Aldrich, St. Louis, MO, USA) was used as received.

### 2.2. CNF Filament Preparation

CNF filaments were prepared via ice-templating using a setup described elsewhere [27]. The procedure consisted of pouring a 0.05 or 0.1 wt% CNF water suspension inside a polytetrafluoroethylene (PTFE) mold and unidirectionally freezing it. The copper bottom plate of the mold was placed on top of a copper rod immersed in liquid nitrogen. The temperature was controlled with a PID-controlled heating element attached to the rod. A cooling rate of 40 °C/h was used in all the experiments. Dry CNF filament mats were obtained by sublimating the ice inside a freeze dryer for approximately 4 days.

Silane-treated CNF filaments were prepared via CVD. A beaker with 1 mL of APTES and the freeze-dried filament mats were placed inside a desiccator. A vacuum pressure of approximately 0.95 bar was applied to the desiccator, which was then kept inside an oven at 150 °C for 1 h. After repressurizing and cooling the setup, CNF filament mats with an APTES surface were obtained.

### 2.3. Preparation of CNF Filament Composites

Composite materials were processed using vacuum infusion [22,27]. Four 0.05 wt% or two 0.1 wt% CNF filament mats were placed on a metal mold and covered with peel ply and breather cloth to facilitate the resin flow into the mat. Sealant tape and a plastic film were used to seal the system, and a vacuum was used to fill the filament mats with a degassed bio-epoxy resin. After filling the whole system with the resin, the outlet and inlet tubes were clamped, and the system was left to cure at room temperature. After 24 h, the samples were demolded, post-cured at 80 °C for 2 h, and left to cool to room temperature. Finally, the samples were polished to remove the surface roughness created by the peel ply.

### 2.4. Characterization

The CNF filaments were imaged with an optical microscope (Leica MZ FL III, Leica Camera AG, Wetzlar, Germany). A piece of the filament mat was placed between glass slides for imaging.

The CNFs, CNF filament mats, and composite fracture surfaces were imaged with FESEM (ZEISS ULTRA Plus FE-SEM, Carl Zeiss AG, Oberkochen, Germany). The CNF sample was carefully collected on a 0.2 µm polycarbonate membrane via vacuum filtration, frozen with liquid nitrogen, and freeze-dried. A piece of the membrane containing the dried CNFs was attached to a sample holder with carbon tape. The CNF filament samples were prepared by gently putting the filament mats in contact with carbon tape attached to sample holders, after which some of the filaments had been glued to the tape. The average diameters of the filaments were calculated by measuring 100 filaments from the SEM images. The composite fracture surfaces were prepared by immersing a piece of the material in liquid nitrogen and breaking it with two tweezers.

TEM (JEOL JEM-2200FS, JEOL Ltd., Tokyo, Japan) was used to image the CNFs. A drop of dilute suspension was applied on a carbon-coated grid and colored with uranyl acetate.

The specific surface area of the CNF mats was determined by N2 physisorption using the Brunauer–Emmett–Teller (BET) method (Micromeritics ASAP 2020). The samples were kept inside an oven at 105 °C for 18 h before testing to remove adsorbed moisture.

Viscoelastic properties of the composites were characterized with DMA (Q800 DMA, TA Instruments, New Castle, DE, USA). Specimens of size 30 × 3 mm^2^ were cut from the vacuum-infused samples, polished to a thickness of 0.1 mm, and tested under tension mode. The span length was set at 15 mm. A displacement of 15 µm and a ramp rate of 2 °C/min from 30 to 150 °C were used. Only the 30–100 °C range is reported as no meaningful changes in the properties were detected at higher temperatures. The results were analyzed using one-way ANOVA and Tukey’s tests with a 0.05 significance level.

DRIFT spectroscopy (Bruker Vertex 80V, Bruker, Billerica, MA, USA) was used to study the chemical composition of the CNF filament mats. The mats were pressed into pellets, and spectra from 400 to 4400 cm^−1^ were obtained. Forty scans with a 4 cm^−1^ resolution were performed for each sample.

XPS (ESCALAB 250Xi, Thermo Fisher Scientific, Loughborough, UK) was used to analyze the surface chemistry of the CNF filaments. A monochromatic Al Kα X-ray source was used at 300 W. The analyzer pass energy was 150 eV for the survey scan and 20 eV for the high-resolution scans.

The alignment of CNFs in the ice-templated CNF filament mats was studied by WAXS measurements on an Anton Paar SAXSpoint 2.0 system (Anton Paar, Graz, Austria) equipped with a Microsource X-ray source (Cu K-alpha radiation, wavelength 0.15418 nm) and a Dectris 2D CMOS Eiger R 1M detector with a 75 × 75 µm^2^ pixel size. All measurements were performed with a beam size of approximately 500 µm diameter and a beam path pressure of about 1–2 mbar. The sample-to-detector distance was 111 mm during the measurements. All samples were mounted on a Sampler for Solids 10 × 10 mm^2^ (Anton Paar, Graz, Austria) holder. Three frames of 24 min duration were read from the detector, giving a total measurement time of 1.2 h per sample. The transmittance was determined and used for scaling of intensities. The software used for instrument control was SAXSdrive version 2.01.224 (Anton Paar, Graz, Austria), and post-acquisition data processing was performed using the SAXSanalysis version 4.00.046 (Anton Paar, Graz, Austria).

The orientation of the cellulose crystals in the composites was studied at the beamline NanoMAX of the MAX IV synchrotron laboratory (Lund, Sweden), and the two-dimensional WAXS patterns were obtained. A photon energy of 10 keV was used. The beam size was 250 × 250 nm^2^, and a sample area of 80 × 40 µm^2^ was analyzed in a single scan. The orientation index (*f*_c_) of the cellulose crystals was calculated according to the intensity distributions of the azimuthal angle using the following equation [31]:(1)$$f_{c}=\frac{180{}^{\circ}-\mathrm{FWHM}}{180{}^{\circ}},$$
where FWHM is the full width at the half-maximum of the azimuthal angle distribution.

## 3. Results and Discussion

### 3.1. CNF Filament Morphology

Figure 3a shows a photograph of an ice-templated CNF filament mat. The material is soft and does not retain the initial shape after demolding, unlike the aerogels reported previously [22,27]. The sample size is approximately 10 × 6 × 2 cm^3^ (length × width × thickness) when inside the mold but changes immediately during and after demolding. This is a result of the open filamentous structure (Figure 3b), which significantly differs from the closed honeycomb structure usually found in CNF aerogels [18,19,22,27]. Due to the low concentration, the CNFs in the water suspension have been arranged by the growing ice crystals into thin strands oriented in the freezing direction instead of forming hexagonal pores. Figure 3c–f shows the morphology of the filaments in detail. The average diameters of the filaments prepared from 0.05 and 0.1 wt% suspensions are 558 ± 186 and 1073 ± 472 nm, respectively, as measured from FESEM images, and the materials are henceforward called 0.56 and 1.1 µm filaments accordingly. The specific surface areas of the 0.56 and 1.1 µm CNF filament mats are 10.68 and 6.89 m^2^/g, corresponding to theoretical average filament diameters of 250 and 387 nm. The difference between the measured and theoretical values might be due to the irregular shape and surface roughness that increase the surface areas of the filaments compared to perfectly smooth cylinders assumed in the theoretical values. In addition, some smaller filaments may have been unintentionally excluded from the manual measurements because they are less visible in FESEM images.

The filaments are similar to those reported elsewhere. For example, Han et al. (2013) obtained oriented filaments with average diameters ranging from 0.57 to 1.5 µm when using a 0.05 wt% nanocellulose water suspension [26]. The diameter of the filaments was determined by the type of nanocellulose used. For cellulose nanocrystals (CNCs), the presence of surface sulfate groups, and thus greater repulsion between the fibers, was suggested to result in thinner filaments. The higher tendency of mechanically fibrillated CNFs to aggregate caused the corresponding filaments to have an average diameter twice as large as that of the nanocrystal-based ones. Meanwhile, Chen et al. (2014) reported diameters ranging from 50 to 300 nm and from 150 to 900 nm for filaments made from high-intensity ultrasonication-induced CNFs prepared with and without prior TEMPO-mediated oxidation, respectively [25]. The authors did not elaborate why the sizes are different, but it can be deduced that the electrostatic repulsion caused by the carboxyl groups on the surface of TEMPO-CNFs, with the additional influence of the reported smaller nanofiber size, resulted in a smaller filament diameter. The size of the filaments reported here falls on the upper end of the size range for similarly prepared materials, agreeing well with the fact that mechanically fibrillated CNFs with no surface functionality were used as a raw material. A reduction in the average diameter could be expected by utilizing, for example, CNCs or TEMPO-CNFs as a raw material.

### 3.2. CNF Orientation in the Filaments

Figure 4 shows a WAXS image together with an identification of the cellulose crystal structure and distribution of orientation for crystalline cellulose. The (200), $\left(1\overset{¯}{1}0\right)$, and (110) reflections can be used to quantify the orientation of both the cellulose crystals and the CNFs since the crystals are aligned in the direction of the nanofibers. By determining the baseline and maximum intensities in the intensity distribution of the azimuthal angle of the $\left(1\overset{¯}{1}0\right)$ and (110) reflections, the orientation index *f*_c_ of the CNFs in the filaments can be calculated according to Equation (1). If all nanofibers are aligned in the same direction, *f*_c_ = 1, and if they are randomly distributed, *f*_c_ = 0. The orientation index for the ice-templated CNF filaments prepared from both 0.05 and 0.1 wt% CNF water suspensions is 0.6, indicating that the nanofibers are partially oriented along the filaments. It should be noted that the calculated orientation index is related to the orientation of the CNFs in a bundle of filaments and is not directly related to the orientation of the microscopic filaments inside the bundle or to the orientation of the CNFs in single filaments.

Orientation index values ranging from approximately 0.56 to 0.82 have been reported for ice-templated nanocellulose networks [18,24]. The value was almost constant at higher fiber contents but dropped dramatically when the suspension concentration was lower than a certain critical value [18]. This critical value was 0.2 wt% for CNCs and 0.08 wt% for CNFs. The drop in the orientation index was suggested to be caused by wrinkling and bending of the fiber structures in the low fiber content networks. Similarly, the filaments reported in the current study are soft, causing the structure to change during demolding and handling of the samples. The orientation of the filaments, and thus also the CNFs, in the dried filament bundles is not the same as the orientation after the freezing process and prior to drying. The dried structure is not fixed but gets easily collapsed, and the filaments do not retain their original alignment, as seen in Figure 3a.

Orientation indices ranging from 0.83 to 0.92 have been reported for CNF filaments prepared via flow focusing [17]. However, the measurements were conducted on single filaments, making the CNF orientation a property of the filament itself and not that of an assembly of filaments as is the case in the current study. The filaments reported here are not readily separable from the network they are a part of, and no information on the nanofiber orientation inside single filaments can be currently provided. Another approach is inducing orientation in cellulose nanopapers by mechanically pulling, or drawing, the wet papers [32]. A highest orientation index of 0.82 was reported for such CNF networks. However, these kinds of networks are difficult to impregnate, as discussed in the introduction, and, unlike the filament mats presented here, are not easily applicable in traditional composite processing.

### 3.3. CNF Filament Surface Characteristics

The CNF filaments were analyzed using DRIFT and XPS to evaluate the CVD process and the resulting aminosilane surface coverage (Figure 5). The DRIFT spectra (Figure 5a) show typical cellulose peaks at around 3380 and 1640 cm^−1^, 2900 and 1250–1460 cm^−1^, and 1050–1170 cm^−1^, which can be attributed to hydroxyl (OH), alkyl (CH_2_), and C–O–C groups, respectively [33,34], and are observed in all samples. On the other hand, the peak at 1570 cm^−1^ can be assigned to N–H bending of the primary amine of the silane molecule [34,35,36,37] and is only present in the silylated samples. This is a clear indication of a successful surface coverage. The potential Si–O bridges are indistinguishable from the C–O–C vibrations, and no conclusions can be made concerning the bonding between the aminosilane and cellulose molecules [16,38]. However, APTES has been shown to form covalent bonds with cellulose when treated at an elevated temperature [39].

Figure 5b shows the XPS spectra of a reference non-silylated filament mat and silylated 0.05 and 0.1 wt% filament mats. All samples show the typical oxygen (O 1s) and carbon (C 1s) peaks at 533 and 286 eV, respectively [40]. Both of the silylated samples show two additional peaks at 399 and 102 eV, corresponding to nitrogen (N 1s) and silicon (Si 2p), respectively [40]. These peaks indicate the presence of the aminosilane chemical on the filament surface, further confirming the results from DRIFT. The reference sample also shows a minor silicon peak, which is most likely caused by a contamination originating from the various processing steps such as the mechanical grinding.

The C 1s peak is divided into several components for all samples. The peaks at 288.1 and 286.6 eV can be ascribed to O–C–O or C=O and C–O bonds, respectively, and are typical of cellulosic materials [16,41]. The peak at 284.8 eV is related to both C–C and C–Si bonds. The relative size of this peak is significantly bigger for the silylated samples, further confirming the presence of the aminosilane molecule, which has a backbone consisting of three carbon atoms and one silicon atom (Table 1).

### 3.4. Composite Morphology

The composite fracture surfaces show the CNF filaments embedded in the epoxy matrix (Figure 6). The side profiles of the freezing-oriented filaments are seen in the longitudinal cross sections (Figure 6a). A significant difference can be observed between the silylated and the non-silylated samples. In the non-silylated composites, the filaments appear to be separate from the matrix with visible gaps between the two components. On the other hand, the silylated filaments are well integrated in the matrix. This is even more pronounced in the transverse cross sections (Figure 6b). The non-silylated samples show a substantial number of fiber pullouts, and there appear to be significant gaps between the filaments and the matrix, indicating poor interfacial adhesion. Similar findings were made in a previous study [27]. The silylated filaments form a more homogenous material with the epoxy matrix, and the breakage has primarily occurred in a brittle fashion in contrast to debonding.

### 3.5. CNF Orientation in the Composites

The orientation of cellulose crystal and the CNFs in the composites were also analyzed, and Figure 7a,b shows a representative 2D-WAXS diffractogram of a 1.1 µm filament/epoxy composite and its corresponding radial integration, respectively. The main peak at 2*θ* of 18.8°, attributed to the presence of epoxy [42], merges with the strongest cellulose peak from (200) planes seen at 22.5°, making it difficult to analyze the CNF alignment. However, the peak from cellulose (004) planes at 34.3° is not affected by epoxy [43]. An azimuthal integration of the (004) planes was carried out, and the resulting curve is shown in Figure 7c. The orientation index calculated from Equation (1) is 0.53, indicating that the CNF alignment is retained after the epoxy impregnation.

The orientation index value is lower than the 0.84 reported for cellulose nanocomposites prepared from CNCs and carboxymethyl cellulose [44]. However, this value is not directly comparable to the results obtained in the current study, as the orientation was induced by drawing a water-based mixture of nanofibers and a polymer matrix. The method is not applicable to thermoset composites. To the best of our knowledge, quantitative data on the orientation of CNFs, or any other type of nanocellulose, in thermoset composites have not been reported before.

### 3.6. Composite Viscoelastic Properties

All of the composites with 18 wt% CNF filament content show improved storage modulus values throughout the 30–100 °C test range (Figure 8, Table 2). The improvement is most significant in the rubbery state due to a continuous fiber network formed by the filaments, a phenomenon widely reported for cellulose nanocomposites [6,9,27,45]. The storage modulus in the longitudinal, i.e., filament, direction is increased up to 2.5-fold compared to that of the epoxy polymer at 30 °C. Thus, the reinforcement effect is more significant than in the previously reported aerogel-based epoxy composites [22,27] and is comparable to those reported for epoxy composites with 20 wt% tunicate whiskers (3.1-fold) [45]. On the other hand, the highest storage modulus value in the transverse direction corresponds to a 1.7-fold increase. All samples show higher values in the longitudinal direction resulting from the freezing process used for preparing the filament mats. The vertically advancing ice front has pushed the suspended CNFs into threadlike filaments that are oriented along the freezing direction. However, it should be emphasized that the difference in the longitudinal and transverse properties is not directly related to the orientation of individual CNFs but to that of the filaments formed by the nanofibers. Only by combining the observations from the WAXS and DMA studies, it can be deduced that both the single CNFs and the filaments formed by the CNFs are at least partially oriented in the composites and that these two things are related to each other.

The silylation does not have a significant effect on the storage moduli. The storage modulus in the transverse direction is higher for the composites with silane-treated CNF filaments, but the difference is not significant, and the longitudinal values are almost identical. However, the silylation has a remarkable effect on the tan delta peak temperature. The peak temperature of the longitudinal specimens is shifted from 72.1 to 80.2 °C for both of the composites with silylated filaments. This indicates good interfacial adhesion between the filaments and the epoxy matrix and is a major improvement compared to the nontreated ones, showing no temperature shift at all. The values are also superior to those reported earlier for CNF aerogel/epoxy composites [22,27] and the increase in the tan delta peak temperature is comparable in scale to that achieved with cellulose whiskers using a solvent casting approach [45].

It is noteworthy that all samples show significant variation between the test specimens, indicating the presence of inhomogeneities in the fiber contents of the composites. The three test specimens were cut out from different parts of the samples and may thus contain different fiber fractions. In addition, the small specimen size and the fact that the samples were not perfectly even due to manual polishing may have contributed to errors in the measured specimen dimensions. Thus, not all of the differences in the DMA results are statistically significant (Table 2).

## 4. Conclusions

Trying to find ways of utilizing renewable raw materials such as cellulose instead of petroleum- or mineral-based ones in everyday applications is becoming more and more important. Composite materials are widely used as load-bearing structures, for example, in transportation, but the traditional reinforcement materials they contain are based on nonrenewable resources. Natural fibers from various plants can be used, but they do not necessarily meet the demands of many modern products. By processing the fibers into the smallest structural units, namely, nanocellulose, the remarkable mechanical and physical properties of the cellulose crystal can be accessed.

A major challenge in using nanocellulose in composites is controlling the micro- and macroscopic architecture and avoiding the fibers from forming agglomerates that negatively affect the properties of the final product. Ice-templating is a fascinating approach to assemble nano- and microscopic particles into macroscopic objects and applying the method to nanocellulose gives rise to lightweight foamlike materials in which the structure is oriented along the freezing direction of the ice crystals. With a low-enough fiber concentration, a network of interconnected thin filaments is formed instead of a honeycomb structure. In theory, the smaller the filament diameter, the better the properties of both the filaments and the composite material containing them. Thus, the possibility of obtaining ultrathin filaments with diameters less than 1 µm is intriguing.

This study shows the applicability of the ice-templating approach in preparing cellulose nanocomposites. Thin filaments were formed from mechanically processed CNFs, and the filament network structures were impregnated with a bio-epoxy resin using vacuum infusion. The resulting composites had an 18 wt% fiber content, and the mechanical properties were significantly improved compared to neat epoxy. The fiber–matrix interface could be optimized by utilizing a simple CVD process to cover the cellulosic surfaces with an aminosilane capable of forming covalent bonds with the epoxy polymer. The effect of the silylation procedure was distinguishable in both the FESEM and DMA studies. The treated filaments were well integrated into the matrix, and the tan delta peak temperature was up to 8 °C higher for the composites with silylated filaments.

## Figures and Tables

**Figure 1 nanomaterials-11-00490-f001:**
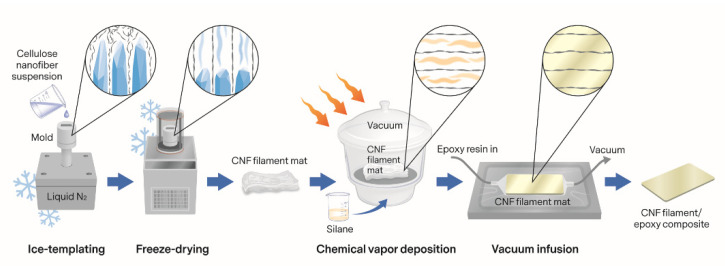
Process schematic. Ice-templating was used to orient the cellulose nanofibers (CNFs) into thin filaments, and dry filament mats were obtained after freeze-drying. The filament surfaces were covered with an aminosilane, and the treated filament mats were impregnated with a bio-epoxy resin to form composites.

**Figure 2 nanomaterials-11-00490-f002:**
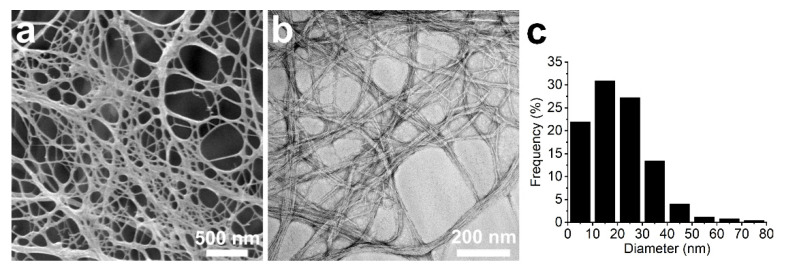
Cellulose nanofibers. (**a**) FESEM and (**b**) TEM images, and (**c**) the size distribution of the CNFs used as a raw material.

**Figure 3 nanomaterials-11-00490-f003:**
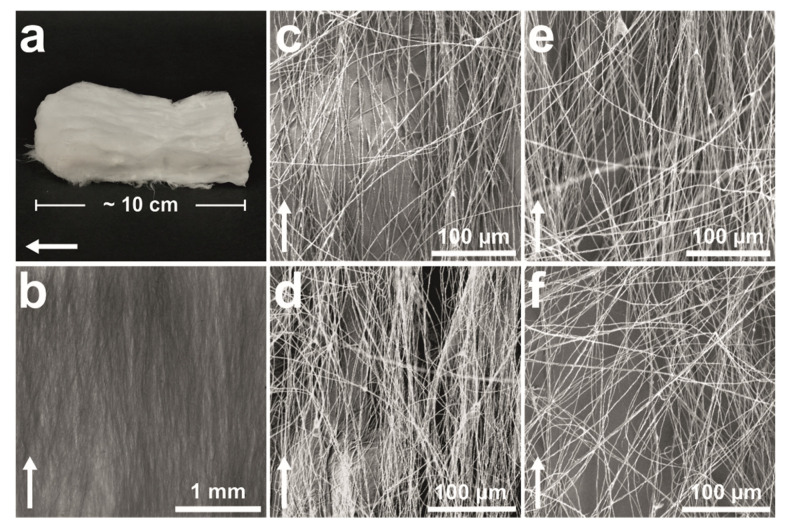
CNF filaments. (**a**) A photograph and (**b**) an optical microscope image of a CNF filament network. FESEM images of non-silylated (**c**) 1.1 and (**d**) 0.56 µm and silylated (**e**) 1.1 and (**f**) 0.56 µm filaments. The white arrow indicates the freezing direction.

**Figure 4 nanomaterials-11-00490-f004:**
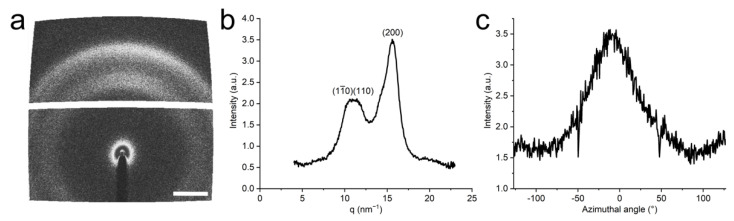
Cellulose crystal orientation in the filaments. (**a**) A representative WAXS diffractogram of a CNF filament in the bundle. The scale bar represents *q* = 5 nm^−1^. Sample direction is set on equatorial direction. Sample and nanofiber directions coincide. (**b**) Radial integration of the diffractogram after background subtraction. (**c**) Azimuthal integration of the $\left(1\overset{¯}{1}0\right)$ and (110) scattering planes, *q* = 11 ± 0.5 nm^−1^. The zero degree of azimuthal angle is set on meridian.

**Figure 5 nanomaterials-11-00490-f005:**
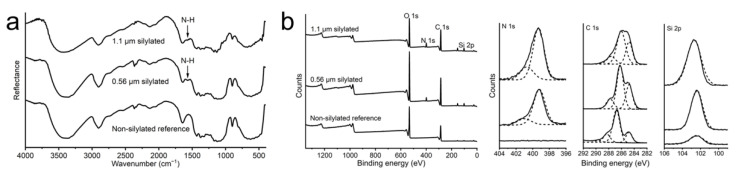
DRIFT and XPS spectra. (**a**) DRIFT and (**b**) XPS spectra of the CNF filament mats.

**Figure 6 nanomaterials-11-00490-f006:**
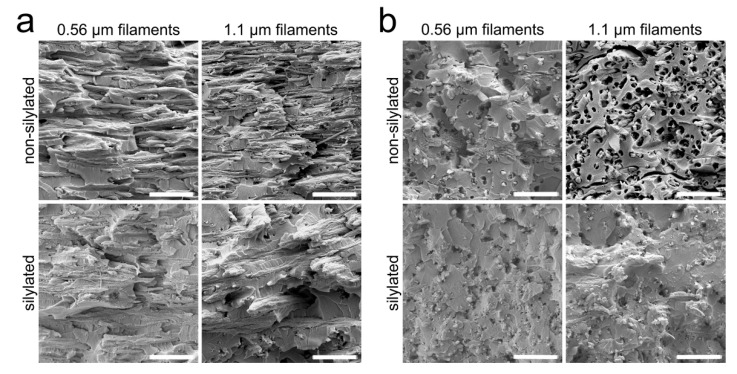
Composite morphology. Longitudinal (**a**) and transverse (**b**) fracture surfaces of the CNF filament/epoxy composites. (The scale bar is 5 µm in all images.)

**Figure 7 nanomaterials-11-00490-f007:**
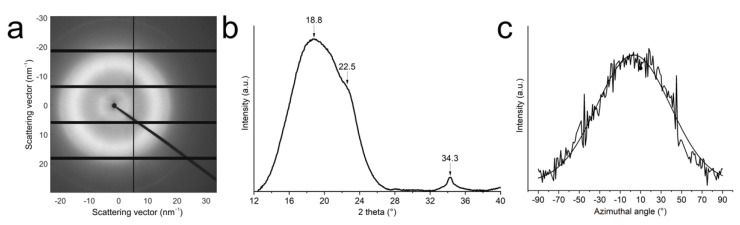
Analysis of filament alignment in the composites using 2D-WAXS. The meridional direction coincides with the freezing direction during ice-templating, i.e., the filament direction. (**a**) A representative diffractogram of a 1.1 µm filament/epoxy composite. (**b**) Radial integration of the diffractogram after baseline subtraction. (**c**) Azimuthal integration of the (004) plane. The zero degree was set on meridian.

**Figure 8 nanomaterials-11-00490-f008:**
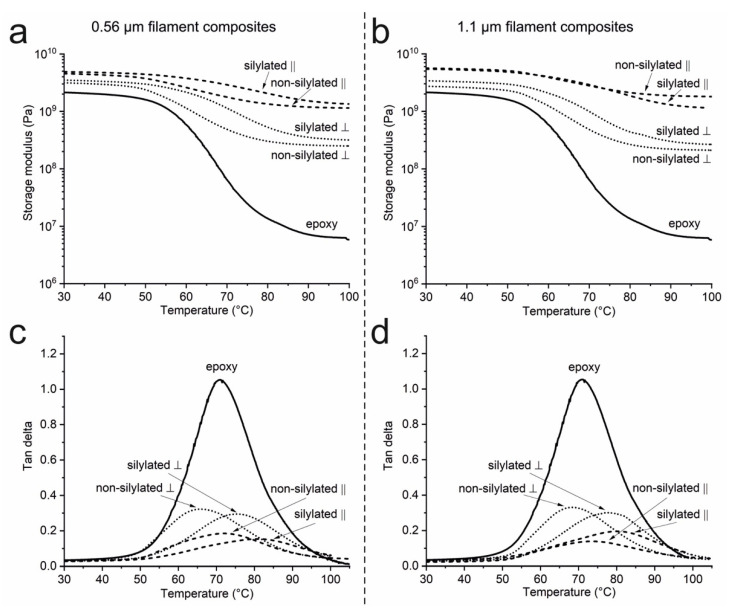
Viscoelastic properties of the CNF filament/epoxy composites. Storage modulus of the (**a**) 0.56 and (**b**) 1.1 µm filament composites. Tan delta of the (**c**) 0.56 and (**d**) 1.1 µm filament composites. Longitudinal (‖) and transverse (┴) specimens were prepared from the composite samples.

**Table 1 nanomaterials-11-00490-t001:** Chemical composition of the CNF filament surfaces. Fractions of elements in percentages (%) on the silylated and non-silylated filament surfaces according to XPS analysis.

Sample	O 1s	C 1s	N 1s	Si 2p	(C–C + C–Si): (O–C–O + C–O)
1.1 µm silylated	30.05	55.31	6.55	7.64	0.42
0.56 µm silylated	40.39	50.85	3.88	4.89	0.44
Non-silylated reference	42.58	55.54	0.00	1.78	0.21

**Table 2 nanomaterials-11-00490-t002:** Viscoelastic properties of the CNF filament/epoxy composites. Longitudinal (‖) and transverse (┴) specimens were prepared from the composite samples.

Sample	Storage Modulus at 30 °C (MPa) *	Storage Modulus at 100 °C (MPa) *	Tan Delta Peak (°C) *
Epoxy	2180 (± 180) ^a^	6.52 (± 0.53) ^a^	72.1 (± 1.0) ^a^
0.56 µm ‖	5030 (± 510) ^b^	1270 (± 230) ^b^	71.3 (± 1.1) ^ab^
0.56 µm silylated ‖	5050 (± 360) ^b^	1250 (± 100) ^b^	80.2 (± 0.4) ^d^
0.56 µm ⊥	3270 (± 170) ^a^	270 (± 64) ^a^	66.5 (± 0.6) ^c^
0.56 µm silylated ⊥	3600 (± 260) ^a^	381 (± 69) ^a^	76.4 (± 0.7) ^e^
1.1 µm ‖	5350 (± 560) ^b^	1440 (± 440) ^b^	73.4 (± 0.5) ^a^
1.1 µm silylated ‖	5260 (± 390) ^b^	1070 (± 170) ^b^	80.2 (± 0.9) ^d^
1.1 µm ⊥	2970 (± 350) ^a^	187 (± 84) ^a^	68.8 (± 1.1) ^bc^
1.1 silylated ⊥	3360 (± 140) ^a^	282 (± 91) ^a^	78.2 (± 0.5) ^de^

* Values with the same superscript (a, b, c, d, and e) within a column do not differ significantly according to one-way analysis of variance (ANOVA) and Tukey’s tests with a 0.05 significance level.

## Data Availability

Data presented in this study are available on request from the corresponding author.

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
