# Peer review of "Ice-Templated Cellulose Nanofiber Filaments as a Reinforcement Material in Epoxy Composites"

_nanomaterials, 2021, doi:10.3390/nano11020490_

Round 1
Reviewer 1 Report
The authors were described that preparation and evaluation of nanocomposite with nanocellulose which prepared by the ice templated method. Ice templated cellulose nanofiber and its mat after modification with APTES were characterized and interface adhesive between epoxy resin and fiber also were evaluated through mechanical and spectral properties. Basically, I agree to publish for Nanomaterials after minor revision.
- The authors prepared CNF filament mat before surface modification with APTES. Don’t you have any knowledge or any reason why surface modified CNF doesn’t use ice-template method?
- The CNF was prepared by themselves. Do you have any information regarding nanofiber production such as that of length, width, shape, crystallinity?
- The author fixed CNF concentrations 0.2wt% and 0.08wt% as that of critical value. In order to affect the mechanical property after the production of composites, have you considered the concentration and crystallinity of CNFs?
- Regarding Table 1, Hove you considered EDX for elemental analysis on the surface?
- CNF mat was modified by APTES having amine group which may be work as an initiator for epoxy-resin. If the author doesn`t use Super Sap INH and CLX hardener, CNF filament composite can be obtained or not?
- Does the author has any proof of filament mat form covalent bonding with epoxy resin?
Author Response
Dear Editor and respected reviewer 1, we are grateful for your comments and corrections
Reviewer 1
Dear reviewer, thank you for your comments. We have done our best to answer your questions point by point:
Comment 1. The authors prepared CNF filament mat before surface modification with APTES. Don’t you have any knowledge or any reason why surface modified CNF doesn’t use ice-template method?
Response 1: Thank you for the comments. There are two reasons for conducting the surface treatment after the ice-templating and not before. First, the CNFs cannot be dried without aggregation, so the surface treatment of the CNF before the ice-templating would require solvent exchange which would be more complicated than the chemical vapor deposition approach used here. Another important reason is that the formation of the filaments from the CNFs is highly dependent on the network-forming and hydrogen bonding of the unmodified CNFs as a result of hydroxyl groups on the CNF surface. The silane treatment prior to the ice-templating would decrease the fiber-fiber interactions and thus the formation of a network structure.
Comment 2. The CNF was prepared by themselves. Do you have any information regarding nanofiber production such as that of length, width, shape, crystallinity?
Response 2: Thank you so much for this question. We have added microscopy images of the CNFs and their size distribution in the manuscript (Figure 2). This raw material after mechanical nanofibrillation has a crystallinity about 68% measured in our earlier study (ref. Hietala et al Waste Management 80 (2018) 319–326). We have now added this information as well as the chemical composition of the pulp.
Comment 3. The author fixed CNF concentrations 0.2 wt% and 0.08 wt% as that of critical value. In order to affect the mechanical property after the production of composites, have you considered the concentration and crystallinity of CNFs?
Response 3: We would like to clarify that the mentioned critical values are not related to the cellulose filament content in the composites. The fiber/filament content in the composites is mainly determined by the network structure and its ability to be compacted by the applied vacuum pressure subjected during the vacuum infusion process.
The mentioned critical values 0.2 wt% for cellulose nanocrystals and 0.08 wt% for cellulose nanofibers are referring to the mechanical integrity of ice-templated nanocellulose structures, explained in the Ref 18. The authors of the reference [18] found that if the fiber concentration in the water suspension used in the ice-templating process reached those values, the resulting fiber networks did not retain their initial shape after drying because of the softness and/or fragility of the structures.
Comment 4. Regarding Table 1, have you considered EDX for elemental analysis on the surface?
Response 4: We are not sure about the reason for this comment, but no, we didn’t consider EDX as surface analysis because EDX is measuring more in depth (several micrometers) while the XPS is on surface (nm depth).
Comment 5. CNF mat was modified by APTES having amine group which may be work as an initiator for epoxy-resin. If the author doesn`t use Super Sap INH and CLX hardener, CNF filament composite can be obtained or not?
Response 5: This is an interesting idea and question. The amino content of the treated filaments is very low and not sufficient for curing, furthermore the filament content is also low. In addition, the use of hardeners decreases the viscosity of the resin significantly making the vacuum infusion process possible in the first place. Regarding your question if the composite can be made without Supesap and hardener, we expect that these filaments can be used to make different types of composites but for the vacuum infusion process it is important to have a resin with low viscosity.
Below a simple calculation of the content: A stoichiometric amount of hardener is needed for full cure of epoxy, and in our case, according to the manufacturer’s data sheet, that meant using a 100:19:19 weight ratio for epoxy and the two hardeners, respectively. In practice, the prepared composite materials, weighing approximately 1.22 g, contained 0.22 g fibers and 1.0 g of epoxy/hardener mixture out of which 0.28 g was hardener. On the other hand, the CVD process only results in a molecular layer of APTES on the filament surface. This amount was not even detectable via weighing the filament mats before and after CVD, which means that the amount was far from the stoichiometric amount and full cure of the epoxy resin would be very unlikely without using the hardeners.
Comment 6. Does the author has any proof of filament mat form covalent bonding with epoxy resin?
Response 6: Covalent linkages between the cellulose filaments and the epoxy resin are likely but not analytically studied in this paper. However, earlier studies have shown that silane chemicals (including APTES) can form covalent bonds with cellulose at elevated temperatures (Ref 29 Abdelmouleh, M., Boufi, S., ben Salah, A., Belgacem, M. N., & Gandini, A. (2002). Interaction of silane coupling agents with cellulose. Langmuir, 18(8), 3203-3208). The treatment in our study was conducted at 150 °C for 1 h, so some degree of covalent bonding between APTES and the CNF filaments is expected.

Reviewer 2 Report
The manuscript from Nissila et al. describes the possible use of ice-templating for the preparation of microsized cellulose fibers to be surface modified, and furtherly considered as reinforcement, by impregnation, of a bioepoxy. The approach considered for the preparation of the fibers, their CDV modification with silane and their impregration are extremely practical and easily reproducible, so the experimental procedures are fully approved. Additionally, even characterization phase and results have been properly conducted, so in my opinion this manuscript deserves to be considered for publication in its present form, I have no concerns on it.
The only things to be modified are:
- caption of Figure 1, that now it is exactly the same of Figure 2
- line 119, to be removed or correctly modified
Author Response
Dear Editor and respected reviewer 2, we are grateful for your comments and corrections
Reviewer 2
The manuscript from Nissila et al. describes the possible use of ice-templating for the preparation of microsized cellulose fibers to be surface modified, and furtherly considered as reinforcement, by impregnation, of a bioepoxy. The approach considered for the preparation of the fibers, their CDV modification with silane and their impregration are extremely practical and easily reproducible, so the experimental procedures are fully approved. Additionally, even characterization phase and results have been properly conducted, so in my opinion this manuscript deserves to be considered for publication in its present form, I have no concerns on it.
The only things to be modified are:
- caption of Figure 1, that now it is exactly the same of Figure 2
- line 119, to be removed or correctly modified
Response: Dear reviewer, thank you so much for your comments and correction of a mistake with the figure caption. We have corrected the Figure 1 caption and added a new Figure 2. The new Figure 2 shows electron microscope images and the size distribution of the used CNFs.
We have corrected the mistake on line 119.
Reviewer 3 Report
Review of nanomaterials-1101200
This manuscript describes the preparation of oriented cellulose nanofibers with the aid of ice crystal growth. The oriented fibers are then embedded into an epoxy resin, and the fiber orientation and mechancical properties measured. Some of the fibers are modified with a silan in a CVD process before embedding. I think the work is of interest and it was made carefully and is presented well.
What is in my opion a bit missing is a clear correlation of the fiber orientation and its influence on the mechanical properties. Finally, the DMA data is related to parallel and perpendicular fiber directions but this would also be possible without the detail WAXS orientation data or?
Further, I would suggest to add simple tensile tests of the specimen parallel and perpendicular to the fiber. This would make a comparison of the reinforcing effect more visible from a practical point of view.
Author Response
Dear Editor and respected reviewer 3, we are grateful for your comments and corrections
Reviewer 3
This manuscript describes the preparation of oriented cellulose nanofibers with the aid of ice crystal growth. The oriented fibers are then embedded into an epoxy resin, and the fiber orientation and mechancical properties measured. Some of the fibers are modified with a silan in a CVD process before embedding. I think the work is of interest and it was made carefully and is presented well.
What is in my opion a bit missing is a clear correlation of the fiber orientation and its influence on the mechanical properties. Finally, the DMA data is related to parallel and perpendicular fiber directions but this would also be possible without the detail WAXS orientation data or?
Further, I would suggest to add simple tensile tests of the specimen parallel and perpendicular to the fiber. This would make a comparison of the reinforcing effect more visible from a practical point of view.
Dear reviewer, we are grateful for your comments and would like to clarify that DMTA and WAXS studies are on different size scales. The DMTA is giving information of the orientation of the filaments in the epoxy resin and WAXS is giving information of the nanocellulose orientation in the filaments.
The WAXS study showed that the individual CNFs forming the filaments are oriented along the filament axis and the DMTA show that the filaments themselves are oriented inside the polymer matrix. Together these studies compose a more in-depth analysis of the anisotropic nature of the material. It is important to keep in mind that the nanometer-scale CNFs used as a raw material and the micrometer-scale filaments in the filament mats are two different things.
We think that the tensile testing is a good idea, but the focus of the current study was more on the quantitative analysis of the fiber orientation via WAXS studies of both the filament structures and the composites. We have actually shown the mechanical properties more in detail in our previous study in which a clear difference between the tensile and flexural properties in the two directions was observed (reference [27]).